# Shifting Outcomes: Superior Functional Recovery in Embolic Stroke of Undetermined Source Compared to Cardioembolic Stroke

**DOI:** 10.3390/neurolint17030035

**Published:** 2025-02-25

**Authors:** Jessica Seetge, Balázs Cséke, Zsófia Nozomi Karádi, Edit Bosnyák, László Szapáry

**Affiliations:** 1Stroke Unit, Department of Neurology, University of Pécs, 7624 Pécs, Hungary; j.seetge@gmx.de (J.S.); karadi.zsofia@pte.hu (Z.N.K.); bosnyak.edit@pte.hu (E.B.); 2Department of Emergency Medicine, University of Pécs, 7624 Pécs, Hungary; cseke.balazs@pte.hu

**Keywords:** functional recovery, 90-day mRS, mRS-shift, embolic stroke of undetermined source, cardioembolic stroke, propensity score matching

## Abstract

Background/Objectives: An embolic stroke of undetermined source (ESUS) is a subtype of ischemic stroke characterized by a non-lacunar infarct in the absence of a clearly identifiable embolic source, despite comprehensive diagnostic evaluation. While ESUS patients are typically younger, have fewer cardiovascular comorbidities, and experience milder strokes than those with cardioembolic strokes (CEs), their functional recovery remains underexplored. Methods: We retrospectively analyzed data from 374 ischemic stroke patients (*n* = 94 ESUS, *n* = 280 CE) admitted to the Department of Neurology, University of Pécs, between February 2023 and September 2024. Functional recovery was assessed using the modified Rankin Scale (mRS). Propensity score matching (PSM) was performed to balance the baseline characteristics, and the mRS-shift was compared between groups. Independent predictors of mRS-shift were identified using Huber regression and extreme gradient boosting (XGBoost). Results: The ESUS patients were significantly younger (60.7 ± 13.8 years vs. 75.1 ± 11.3 years, *p* < 0.001), had lower pre-morbid modified Rankin Scale (pre-mRS) scores (0.34 ± 0.91 vs. 0.81 ± 1.23, *p* < 0.001), were less likely to have hypertension (75.5% vs. 86.1%, *p* = 0.027) and diabetes (23.4% vs. 36.8%, *p* = 0.024), and presented with milder strokes (National Institutes of Health Stroke Scale [NIHSS] score at admission: 5.4 ± 4.5 vs. 8.1 ± 6.3, *p* < 0.001, and 72 h post-stroke: 3.0 ± 4.4 vs. 6.5 ± 6.3, *p* < 0.001) compared to the CE patients. After adjusting for baseline differences, the ESUS patients demonstrated significantly greater functional recovery than the CE patients (adjusted mRS-shift: 1.84 ± 1.14 vs. 2.53 ± 1.69, *p* = 0.022). Age, pre-mRS score, and NIHSS score at 72 h post-stroke were the strongest predictors of mRS-shift, with an older age, a higher pre-mRS score, and a greater stroke severity significantly decreasing the odds of recovery. Conclusions: The ESUS patients showed superior functional recovery compared to the CE patients, even after accounting for baseline differences. These findings highlight the need for further research into the pathomechanisms underlying ESUSs and the development of optimal treatment strategies to improve patient outcomes.

## 1. Introduction

Stroke remains one of the leading causes of disability worldwide, with ischemic strokes accounting for approximately 87% of all cases [1]. Within ischemic strokes, embolic stroke of undetermined source (ESUS) and cardioembolic stroke (CE) represent two distinct subtypes that share similar underlying mechanisms, but differ in their identifiable causes and clinical management. CE is commonly linked to well-established embolic sources, such as atrial fibrillation (AF), allowing for targeted preventative measures, most notably anticoagulation therapy [2]. In contrast, ESUS, accounting for nearly 17% of all ischemic strokes, presents a diagnostic and therapeutic challenge due to the absence of a clearly identifiable embolic source, despite comprehensive diagnostic evaluation [3].

Emerging evidence suggests that ESUS patients differ significantly from their CE counterparts in terms of baseline characteristics. ESUS patients are generally younger, have fewer cardiovascular comorbidities [4], and typically present with milder neurological deficits upon admission [5,6,7]. Despite these differences, both ESUS and CE patients share similar clinical and radiological features [8], making them comparable groups for evaluating differences in functional recovery.

Previous research has primarily focused on the risk of stroke recurrence and secondary prevention strategies in ESUS patients. Notably, large randomized clinical trials, including NAVIGATE-ESUS (rivaroxaban 15 mg daily with aspirin 100 mg daily), RE-SPECT-ESUS (dabigatran 150 mg or 110 mg daily for patients aged ≥75 years and/or with creatinine clearance 30 to <50 mL/min, compared to aspirin 100 mg daily), ARCADIA (apixaban 5 mg or 2.5 mg twice daily compared to aspirin 81 mg daily), and ATTICUS (apixaban 5 mg twice daily compared to aspirin 100 mg daily) [9,10,11,12], aimed to evaluate the efficacy of anticoagulation in reducing recurrent strokes in ESUS patients, but failed to demonstrate a significant advantage over antiplatelet therapy. Consequently, the optimal secondary prevention strategies for ESUS patients remain controversial.

Despite the growing recognition of ESUS as a distinct stroke subtype, the research focusing on the long-term functional recovery of ESUS patients compared to those with CE remains limited. A more comprehensive understanding of these recovery patterns could guide the development of personalized rehabilitation programs and improve long-term care strategies for ESUS patients.

Moreover, traditional statistical models have limitations in capturing the complex, multifactorial nature of stroke recovery. Recent advances in data analytics, such as machine learning algorithms like extreme gradient boosting (XGBoost), offer more powerful and flexible methods for identifying nuanced predictors of recovery outcomes. XGBoost excels at handling high-dimensional data and uncovering intricate relationships between clinical variables, providing deeper insights into the factors that influence patient recovery.

Therefore, the primary objective of this study is to compare the functional recovery at 90 days between ESUS and CE patients while accounting for baseline differences. By combining traditional statistical models and advanced machine learning approaches, this study aims to identify the critical predictors of functional recovery and deepen understanding of the factors influencing outcomes in ESUS patients.

## 2. Materials and Methods

### 2.1. Study Design and Patient Population

This retrospective study utilized data from the Transzlációs Idegtudományi Nemzeti Laboratórium (TINL) STROKE-registry, which included patients with acute ischemic stroke (AIS) admitted to the Department of Neurology, University of Pécs, between February 2023 and September 2024.

Adult patients (≥18 years) with imaging-confirmed non-lacunar AIS, classified as either ESUS or CE following a comprehensive diagnostic work-up, were included in the study. Patients with incomplete diagnostic evaluations or missing follow-up data were excluded from the analysis.

The study was approved by the local Ethics Committee, and informed consent was waived due to the retrospective nature of the TINL STROKE-registry.

### 2.2. Definitions

ESUS was defined based on the Cryptogenic Stroke/ESUS International Working Group criteria as a visible non-lacunar infarct in the absence of extracranial or intracranial atherosclerosis causing ≥50% luminal stenosis in the arteries supplying the area of ischemia, detected by computed tomography angiography (CTA), magnetic resonance angiography (MRA), or carotid ultrasonography, a major-risk cardioembolic source (excluded through transthoracic echocardiography [TTE] and at least 24 h of cardiac Holter monitoring), and any other specific cause of stroke (e.g., arterial dissection, arteritis, migraine/vasospasm, or drug abuse) [13]. Lacunar stroke was defined as a subcortical infarct ≤1.5 cm (or ≤2.0 cm on magnetic resonance tomography [MRT] diffusion-weighted imaging) located in regions supplied by small penetrating cerebral arteries [13].

CE was defined as a stroke caused by emboli originating from confirmed major cardiac sources (e.g., permanent or paroxysmal AF, intracardiac thrombus) according to the ASCOD phenotyping classification system [14].

### 2.3. Data Collection

Baseline characteristics included the demographics (age, sex), medical history (pre-stroke functional status assessed by the pre-morbid modified Rankin Scale [pre-mRS] score, hypertension, diabetes mellitus), cardiovascular risk factors (current smoking, alcohol consumption), and current medications (e.g., anticoagulation therapy). Stroke-specific data included the interval between stroke onset and hospital admission (onset-to-door time), stroke severity assessed using the National Institutes of Health Stroke Scale (NIHSS) [15] at admission and at 72 h post-stroke, laboratory values (plasma glucose levels and bedside international normalized ratio [INR] at admission), and treatment modalities (standard care [SC], thrombolysis [TL], mechanical thrombectomy [MT], or combined therapy (TL + MT)).

As part of the initial diagnostic work-up, all patients underwent a non-contrast computed tomography (NCCT) scan, CTA/MRA or carotid duplex ultrasonography, a 12-lead electrocardiogram (ECG), TTE, 24 h Holter monitoring, and routine blood tests including the coagulation profiles and vasculitis markers. In selected cases, additional evaluations such as CT perfusion (CTP), transesophageal echocardiography (TEE) with or without a bubble study, extended Holter monitoring, and assessment for prothrombotic conditions, including genetic testing for hypercoagulability, were performed.

### 2.4. Outcome Measure

Functional recovery was assessed by trained neurology staff using the modified Rankin Scale (mRS) score. A favorable functional outcome was defined as an mRS score of 0–2, while an unfavorable outcome was defined as dependency or death (an mRS score of 3–6). Additionally, the mRS-shift was used to evaluate changes in functional status and was defined as the absolute difference between the pre-mRS and 90-day mRS.

### 2.5. Statistical Analyses

Statistical analyses were conducted to compare the baseline characteristics and functional outcomes between the ESUS and CE patients. Continuous variables were reported as means ± standard deviation (SD) and compared using independent *t*-tests (or Mann–Whitney U tests for non-normally distributed data). Categorical variables were analyzed using chi-square (χ^2^) tests or Fisher’s Exact tests when appropriate.

First, we compared the proportion of patients achieving a favorable outcome vs. an unfavorable outcome between the ESUS and CE groups using χ^2^ or Fisher’s Exact tests. To account for confounding variables, ordinary least squares (OLS) regression was applied to adjust the mRS-shift for the baseline characteristics.

To minimize selection bias, 1:1 nearest neighbor propensity score matching (PSM) without replacement was performed. Covariate balance was assessed using standardized mean differences (SMDs) and statistical testing before and after matching. In the matched cohort, Mann–Whitney U tests were used to compare the absolute and adjusted mRS-shift between the ESUS and CE patients.

Subgroup analyses were conducted only in the un-matched cohort; differences in the adjusted mRS-shift between anticoagulated and non-anticoagulated patients were analyzed within the ESUS and CE groups. Similarly, the adjusted mRS-shift was compared between the treatment subgroups (SC, TL, MT, TL + MT).

Huber regression was applied to identify the independent predictors of mRS-shift. This robust regression method reduces the influence of outliers by adjusting the loss function, making it less sensitive to extreme values while maintaining the interpretability of linear regression. Additionally, XGBoost modeling was used to assess predictive factors. XGBoost is a machine learning algorithm based on gradient boosting, which sequentially improves weak models to optimize predictive performance. Hyperparameter tuning was performed to select the best model configuration, and k-fold cross-validation was used to prevent overfitting by training and validating the model across multiple data subsets. SHapley Additive exPlanations (SHAPs) values were computed to quantify the relative contribution of each predictor.

The results were reported as odds ratios (ORs) with 95% confidence intervals (CIs), with *p* < 0.05 considered statistically significant. All statistical analyses were conducted using Python (version 3.13) and R (version 4.4.2).

## 3. Results

This retrospective cohort study analyzed data from 914 AIS patients enrolled in the TINL STROKE-registry at the Department of Neurology, University of Pécs, between February 2023 and September 2024. Of these, 317 patients (34.7%) were diagnosed with CE and 235 patients (25.7%) were classified as having cryptogenic stroke.

In the CE group, 37 (11.7%) were excluded due to missing 90-day mRS scores, leaving 280 CE patients for the final analysis. In the suspected cryptogenic stroke group, 11 patients (4.7%) died before completing the diagnostic evaluation. Among the remaining 224 patients, 98 (43.8%) met the criteria for ESUS. However, four of these (4.1%) lacked 90-day mRS scores and were excluded, resulting in ninety-four ESUS patients included in the final cohort. (Figure 1).

### 3.1. Baseline Differences (Before Matching)

The study cohort consisted of 374 patients, of whom 172 were male (46.0%), with a mean age of 71.51 ± 13.47 years. The ESUS patients were significantly younger than the CE patients (60.7 ± 13.8 years vs. 75.1 ± 11.3 years, *p* < 0.001), had lower pre-mRS scores (0.34 ± 0.91 vs. 0.81 ± 1.23, *p* < 0.001), and were less likely to have hypertension (75.5% vs. 86.1%, *p* = 0.027) and diabetes mellitus (23.4% vs. 36.8%, *p* = 0.024). Additionally, fewer ESUS patients were on anticoagulation therapy at the time of their stroke (7.5% vs. 36.4%, *p* < 0.001).

In terms of the stroke severity, the ESUS patients presented with significantly lower NIHSS scores at admission (5.4 ± 4.5 vs. 8.1 ± 6.3, *p* < 0.001) and at 72 h post-stroke (3.0 ± 4.4 vs. 6.5 ± 6.3, *p* < 0.001), indicating milder strokes.

Furthermore, the ESUS patients were significantly more likely to receive TL than the CE patients (42.6% vs. 19.3%, *p* < 0.001), whereas MT was more frequently performed in the CE patients (23.2% vs. 11.7%, *p* = 0.024). This difference was primarily driven by the higher prevalence of large vessel occlusion (LVO) in the CE group compared to the ESUS group (50.7% vs. 26.6%, *p* < 0.001). However, among the patients with LVO, the distribution of the occlusion sites was similar between the two groups.

The CTP analysis revealed that the CE group (*n* = 89) exhibited larger infarct cores (relative cerebral blood flow [rCBF] < 30%), more widespread hypoperfusion (time-to-maximum delay [Tmax] > 6 s), and a greater mis-match (mis-match ratio [MMR]) compared to the ESUS group (*n* = 26), which showed lower values across these parameters. These findings suggest a higher degree of ischemic damage in the CE patients. However, given the small sample size and the high prevalence of zero values in the ESUS group, these results should be interpreted with caution. A summary of the baseline characteristics, clinical presentation, treatment modalities, and imaging findings for both groups is provided in Table 1.

### 3.2. Functional Recovery (Un-Matched Cohort)

A total of 69.2% of the ESUS patients achieved a favorable functional outcome (mRS score: 0–2), significantly higher than the 41.4% observed in the CE patients (OR = 3.17, 95% CI: 1.93–5.21, *p* < 0.001). Conversely, moderate to severe disability or death (mRS 3–6) occurred in only 30.8% of the ESUS patients, compared to 58.6% in the CE group (Figure 2).

### 3.3. Presumed Underlying ESUS Etiology

Considering the superior functional outcomes observed in the ESUS patients compared to those with CE, we conducted a further investigation into the potential pathophysiological mechanisms that may contribute to these differences. Within our cohort, a variety of presumed embolic sources were identified in the ESUS group, which we categorized into coagulopathies, hypercoagulable states, and paradoxical embolism. Among the hypercoagulable conditions, antiphospholipid syndrome emerged as the most prevalent (*n* = 13), followed by cancer-associated thrombosis (*n* = 6), with the other contributing factors including genetic conditions such as thrombophilia (*n* = 6) and autoimmune diseases including systemic lupus erythematosus (*n* = 1). Additionally, paradoxical embolism resulting from right-to-left shunting via a patent foramen ovale (PFO) was confirmed in nine cases. However, in 59 patients, no clear embolic source or underlying risk factor could be reasonably attributed.

### 3.4. Matching Process and Covariate Balance

To minimize the baseline differences and potential selection bias, PSM was performed using 1:1 nearest neighbor matching without replacement, resulting in a matched cohort of 94 ESUS and 94 CE patients. The SMDs and *p*-values before and after matching were used to assess the balance across key covariates.

Prior to matching, several covariates exhibited substantial imbalance, particularly the age (SMD = −1.15), pre-mRS score (SMD = −0.44), NIHSS score (SMD = −0.48), and 72hNIHSS score (SMD = −0.51). After matching, all SMD values were markedly reduced, with most covariates achieving balance (SMD < 0.25), reflecting improved comparability between the groups (Table 2).

The pre-mRS exhibited a slight residual imbalance post-matching (SMD = −0.26), but statistical testing indicated no significant difference (*p* = 0.072). Additionally, the NIHSS at admission, though still imbalanced after matching (SMD = −0.33, *p* = 0.027), was later found to be a non-significant predictor of mRS-shift in our regression analyses. This suggests that any remaining imbalance had minimal impact on the functional outcomes.

Since slight residual imbalances remained, we additionally examined the adjusted mRS-shift, which was calculated using the OLS regression coefficients obtained previously. This provided an additional layer of control for any potential lingering baseline differences, ensuring that the observed functional recovery differences reflected a true effect rather than residual confounding.

### 3.5. Functional Recovery (Matched Cohort)

After balancing the key baseline characteristics using PSM, the ESUS patients exhibited a significantly lower absolute mRS-shift compared to the CE patients (1.15 ± 1.53 vs. 1.80 ± 1.78, *p* = 0.014). To further account for any remaining baseline imbalances, we assessed the adjusted mRS-shift, which continued to show a significant difference favoring ESUS (1.84 ± 1.14 vs. 2.53 ± 1.69, *p* = 0.022) (Figure 3).

### 3.6. Subgroup Analyses (Un-Matched Cohort)

To further explore the factors influencing functional recovery in the ESUS and CE patients, we conducted subgroup analyses based on the anticoagulation status and treatment modality.

#### 3.6.1. Anticoagulation Status

In the CE cohort, 70 patients were receiving direct oral anticoagulants (DOACs), specifically apixaban (*n* = 49), rivaroxaban (*n* = 9), dabigatran (*n* = 8), and edoxaban (*n* = 4), while 32 patients were on vitamin K antagonists (VKAs), including acenocoumarol (*n* = 19) and warfarin (*n* = 13). In contrast, in the ESUS cohort, only seven patients were on anticoagulation therapy at the time of their stroke, with three on apixaban and four on VKAs (acenocoumarol [*n* = 3] and warfarin [*n* = 1]).

There was no statistically significant difference in functional recovery between the non-anticoagulated and anticoagulated CE patients (adjusted mRS-shift: 1.98 ± 1.44 vs. 2.26 ± 1.76, *p* = 0.391). Similarly, prior anticoagulation had no significant impact on functional recovery in the ESUS patients (adjusted mRS-shift: 1.79 ± 1.11 vs. 2.47 ± 1.43, *p* = 0.215).

#### 3.6.2. Treatment Modalities

Across all treatment modalities, there were no significant differences in the adjusted mRS-shifts between the ESUS and CE patients. Specifically, the adjusted mRS-shifts were as follows: SC: 1.75 ± 1.12 vs. 2.19 ± 1.66 (*p* = 0.267), TL: 1.78 ± 1.11 vs. 1.80 ± 1.46 (*p* = 0.606), MT: 2.57 ± 1.33 vs. 2.24 ± 1.54 (*p* = 0.460), TL + MT: 1.37 ± 0.55 vs. 1.74 ± 1.35 (*p* = 0.900).

### 3.7. Predictors of Functional Recovery (Un-Matched Cohort)

#### 3.7.1. Huber Robust Regression and XGBoost Model

Both Huber regression and XGBoost analysis identified age, pre-mRS score, and NIHSS score 72 h post-stroke as the key predictors of mRS-shift (Table 3, Figure 4 and Figure 5). Increasing age was strongly associated with worse recovery, with each additional year increasing the likelihood of a greater mRS-shift by 69.5% (*p* < 0.001). Similarly, higher pre-mRS scores were linked to poorer functional outcomes, indicating that the patients with greater pre-stroke disability had significantly lower odds of recovery (*p* < 0.001). Stroke severity, assessed by the NIHSS score, was a major determinant of recovery: each additional point at 72 h post-stroke was associated with a 206% decrease in recovery odds (*p* < 0.001). A SHAPs analysis from the XGBoost model confirmed these findings, ranking the 72 h NIHSS score as the most influential predictor, followed by the age and pre-mRS score.

#### 3.7.2. Model Performance

The Huber robust regression model demonstrated good explanatory power (R^2^ = 0.4722) and a strong overall fit (χ^2^ = 358.11, degrees of freedom [df] = 17, *p* < 0.001), confirming that at least one predictor meaningfully contributes to explaining the functional recovery. Complementing this, the XGBoost model, optimized through hyperparameter tuning and validated via k-fold cross-validation, achieved an R^2^ of 0.517 on the test set (*p* < 0.001). This indicates that the model explains 51.7% of the variability in mRS-shift in unseen data, confirming its generalizability. The SHAPs analysis further highlighted the relative importance of the key predictors, reinforcing the model’s strong predictive performance in identifying the factors influencing functional recovery.

## 4. Discussion

This study demonstrated that the ESUS patients experienced significantly better functional recovery than the CE patients, even after adjusting for the baseline differences. While the ESUS patients generally presented with more favorable baseline characteristics, these factors alone do not fully account for their better recovery.

Consistent with the existing literature, the ESUS patients in our cohort were younger, had lower pre-stroke disability and fewer cardiovascular comorbidities, and experienced less severe strokes [5,16,17,18]. Our findings align with previous studies reporting more favorable functional outcomes in ESUS patients compared to CE patients [5,19,20,21,22]. However, some research has reported comparable outcomes between ESUS and CE [23], suggesting that the recovery advantage in ESUS may not be uniform across all populations. Notably, in our cohort, only 30.8% of the ESUS patients had poor outcomes at 90 days (mRS score 3–6), which is within the range of the previously reported rates of 17.5% [24] to 33.2% [25].

### 4.1. Impact of Age on Neuroplasticity and Recovery

The younger age of ESUS patients likely contributes to their better recovery. Age significantly impacts neuroplasticity, the brain’s ability to reorganize and recover after injury. Younger patients have a greater intrinsic capacity for neuroplasticity, which facilitates more effective recovery of lost functions. Additionally, they are generally more physically capable of participating in intensive rehabilitation programs, unlike older CE patients, who often have multiple comorbidities that may limit rehabilitation efforts.

While exercise and rehabilitation strategies can further enhance neuroplasticity, research suggests that aging may limit the extent of functional reorganization [26]. Studies have shown that while rehabilitation improves motor function in both younger and older individuals, the degree of cortical reorganization is more pronounced in younger patients [27]. This may indicate that exercise alone is insufficient for optimal recovery in older stroke survivors.

Beyond traditional rehabilitation, enriched environments—which incorporate cognitive, social, and interactive elements—have been explored as an additional strategy to support neuroplasticity. A 2012 study introduced an enriched rehabilitation setting that included Nintendo game consoles, puzzles, books, and communal spaces for social interaction, allowing patients to engage in activities suited to their individual interests [28]. Since younger patients are generally more accustomed to and engaged with technology-based or interactive activities, such environments may be more easily implemented for them as part of rehabilitation programs.

### 4.2. Stroke Pathophysiology and Infarct Patterns

The differences in functional recovery between ESUS and CE patients may be partly attributed to stroke pathophysiology. ESUS often arises from minor-risk embolic sources, such as a PFO, leading to smaller, localized infarcts [13]. In contrast, CE are typically due to high-risk emboli (e.g., from AF), causing larger infarcts that inflict more extensive damage to critical brain regions like the motor cortex, resulting in poorer long-term outcomes [29].

CTP imaging revealed that the CE patients exhibited larger infarct cores (rCBF < 30%), more widespread hypoperfusion (Tmax > 6 s), and a greater MMR compared to the ESUS patients. In contrast, the majority of the ESUS patients had no significant ischemic core, and more than half had an MMR of 0, suggesting that substantial perfusion deficits were less common in this group. These findings support the hypothesis that ESUS may often result from smaller embolic events or transient hypoperfusion, which standard CTP imaging may not always capture. However, given the small sample size and the high prevalence of zero values in the ESUS group, these results should be interpreted with caution.

Additionally, LVO was significantly more frequent in the CE patients compared to the ESUS patients. However, among the patients with LVO, the distribution of the occlusion sites did not significantly differ between the groups, aligning with previous findings that ESUS can involve multiple vascular territories, similar to CE.

### 4.3. Complexity of Embolic Sources and Treatment Implications

ESUS represents a heterogeneous stroke entity with multiple potential embolic sources [30,31]. In our cohort, the most frequently suspected embolic sources were conditions associated with hyperviscosity, including antiphospholipid syndrome, cancer-associated thrombosis, and inherited thrombophilias. Additionally, PFO-mediated paradoxical embolism was frequently observed, underscoring the heterogeneous pathophysiology of ESUS. However, in the majority of the patients, no embolic source could be suspected, nor could an underlying risk factor be identified, highlighting the persistent diagnostic uncertainty in some cases.

This variability in embolic sources likely influences thrombogenesis and treatment responses. Minor-risk sources like cancer or PFO often lead to red thrombi, which may respond well to anticoagulation, whereas white thrombi from sources like aortic arch or intracranial atherosclerosis tend to be more responsive to antiplatelet therapy [32]. This diversity complicates the optimal secondary prevention strategy for ESUS, as treatment decisions should ideally be tailored based on the underlying embolic source.

While CE patients with AF are routinely managed with anticoagulation, identifying embolic sources in ESUS remains challenging. Since patients initially classified as ESUS with newly detected AF are reclassified as CE, and the AF detection rates in cryptogenic stroke range from 7.6% to 33% depending on the monitoring duration and method [33,34,35], prolonged cardiac monitoring is essential for accurate classification. Additionally, one study found that 29% of ESUS patients were diagnosed with AF over a follow-up period of 3.2 years [4], suggesting that either the AF was initially undiagnosed or that these patients have a higher likelihood of developing AF over time. This highlights the need for not only early prolonged monitoring, but also ongoing follow-up assessments to optimize stroke prevention.

### 4.4. Limitations and Future Directions

Despite offering valuable insights, this study has several limitations. First, the relatively small ESUS sample size (*n* = 94) limits the generalizability and reduces the statistical power, particularly in the subgroup analyses. Second, the retrospective, single-center design introduces potential selection bias, despite the rigorous statistical adjustments. Third, we excluded cryptogenic stroke patients who either died before completing a diagnostic work-up (*n* = 11) or had missing 90-day mRS scores (*n* = 41), which may have introduced additional bias.

Furthermore, the study’s focus on 90-day outcomes may not fully capture the long-term functional recovery, underscoring the need for longer follow-ups to assess sustained recovery and stroke recurrence. Moreover, the study focused solely on functional outcomes and did not assess patient-reported outcomes or quality of life, which are important indicators of recovery and well-being.

Therefore, these findings should be considered hypothesis-generating. Future research should involve larger, multi-center, prospective studies with extended follow-ups to validate and expand upon these results.

## 5. Conclusions

ESUS patients demonstrated superior functional recovery compared to CE patients, even after adjusting for baseline differences. These findings highlight the need for further research into the distinct pathophysiology of ESUS and the development of optimal treatment strategies, considering its diverse underlying mechanisms and implications for personalized stroke care.

## Figures and Tables

**Figure 1 neurolint-17-00035-f001:**
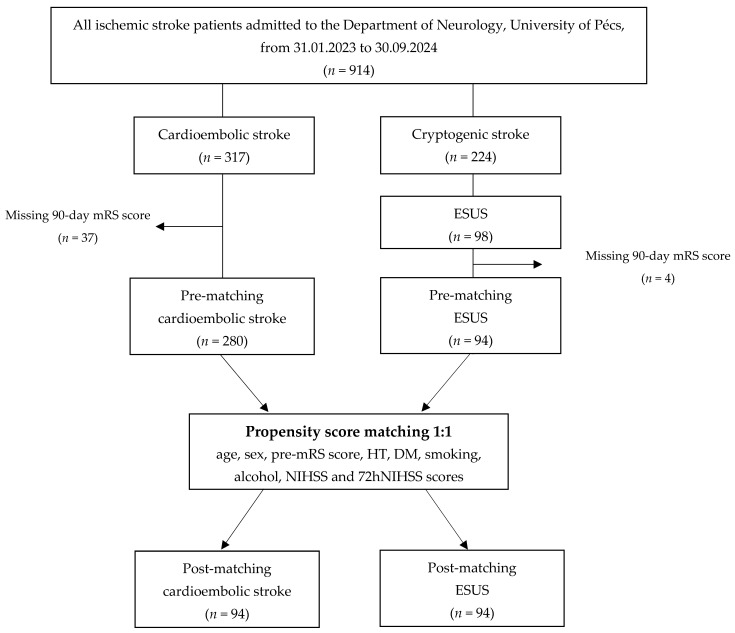
A flowchart of the study. Abbreviations: mRS = modified Rankin Scale, ESUS = embolic stroke of undetermined source, pre-mRS = pre-morbid modified Rankin Scale, HT = hypertension, DM = diabetes mellitus, smoking = current smoking, alcohol = alcohol consumption, NIHSS = National Institutes of Health Stroke Scale at admission, 72hNIHSS = National Institutes of Health Stroke Scale 72 h post-stroke.

**Figure 2 neurolint-17-00035-f002:**
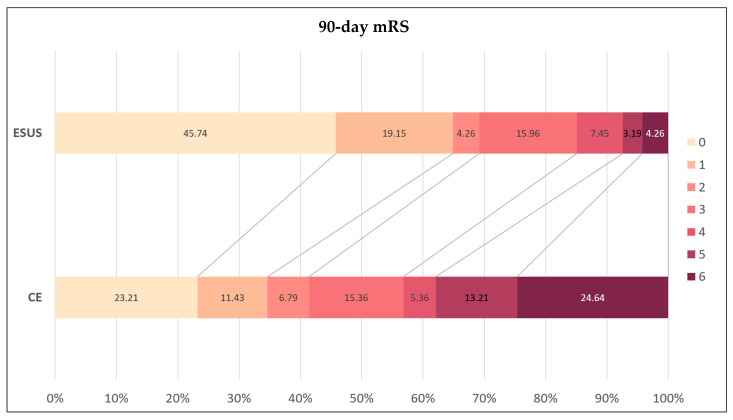
Distribution of 90-day mRS. Abbreviations: mRS = modified Rankin Scale, ESUS = embolic stroke of undetermined source, CE = cardioembolic stroke.

**Figure 3 neurolint-17-00035-f003:**
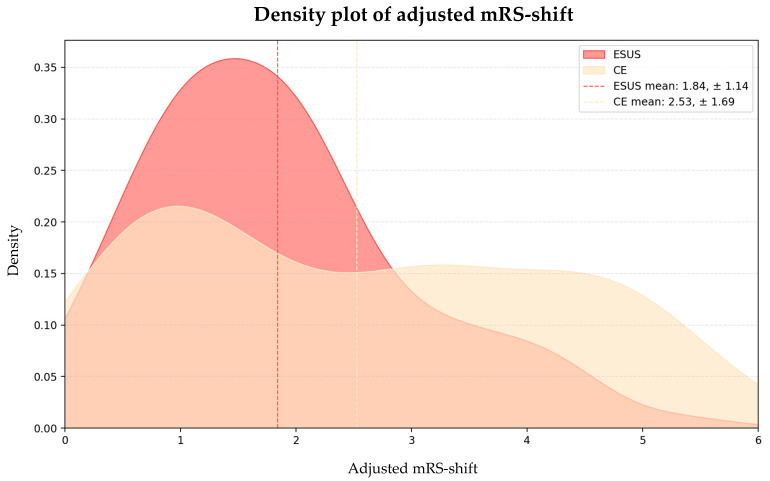
Density plot of adjusted mRS-shift. Abbreviations: mRS = modified Rankin Scale, ESUS = embolic stroke of undetermined source, CE = cardioembolic stroke.

**Figure 4 neurolint-17-00035-f004:**
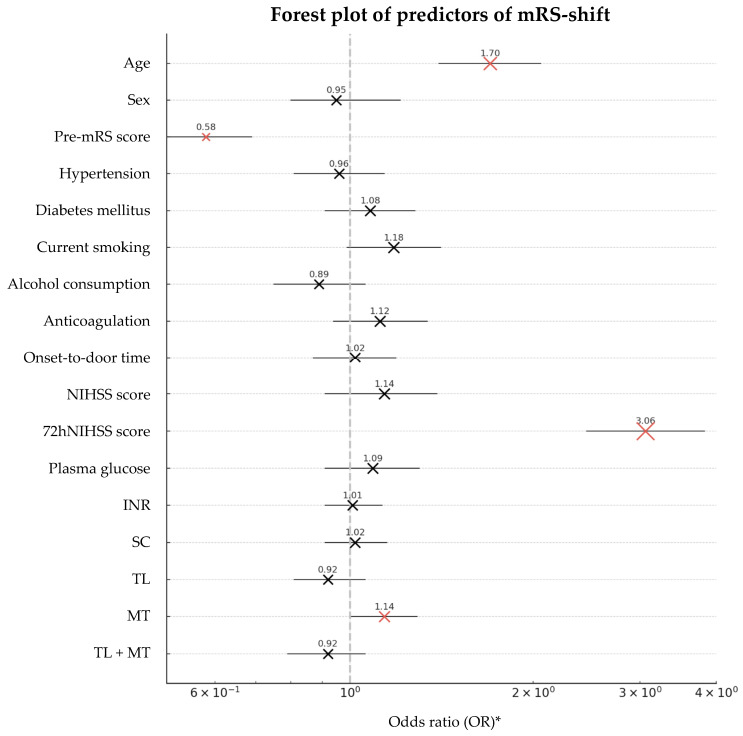
A forest plot of the predictors of mRS-shift. Abbreviations: mRS = modified Rankin Scale, pre-mRS = pre-morbid modified Rankin Scale, NIHSS = National Institutes of Health Stroke Scale at admission, 72hNIHSS = National Institutes of Health Stroke Scale 72 h post-stroke, INR = international normalized ratio, SC = standard care, TL = thrombolysis, MT = mechanical thrombectomy, OR = odds ratio. * The odds ratios are displayed on a logarithmic scale to enhance the visualization of both small and large confidence intervals. Note: The predictors with statistically significant effects (*p* < 0.05) are highlighted in red; MT, while not reaching statistical significance, demonstrates a strong trend in this direction. The size of each marker (‘X’) is proportional to the odds ratio, providing a visual representation of the effect size.

**Figure 5 neurolint-17-00035-f005:**
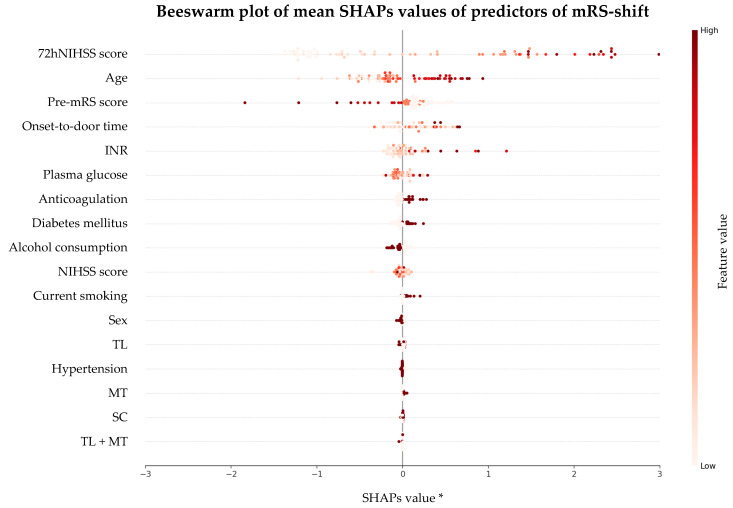
A beeswarm plot of the mean SHAPs values of the predictors of mRS-shift. Abbreviations: SHAPs = SHapley Additive exPlanations, mRS = modified Rankin Scale, 72hNIHSS = National Institutes of Health Stroke Scale 72 h post-stroke, pre-mRS = pre-morbid modified Rankin Scale, INR = international normalized ratio, NIHSS = National Institutes of Health Stroke Scale at admission, TL = thrombolysis, MT = mechanical thrombectomy, SC = standard care. * The range was adjusted for visual consistency. Note: Each dot represents an observation from the test set, with the color indicating the feature value (darker = higher). The SHAPs values on the x-axis show the contribution of each predictor to the model’s prediction of the mRS-shift. The spread of the dots along the x-axis highlights the variability in feature impact across different patients.

**Table 1 neurolint-17-00035-t001:** Baseline differences between ESUS and CE patients.

	ESUS *n* = 94	CE *n* = 280	*p*-Value
Baseline Characteristics			
Age, years, mean (SD)	60.7 (±13.8)	75.1 (±11.3)	<0.001 *
Sex, male, *n* (%)	42 (44.7%)	130 (46.4%)	0.861
Pre-mRS score, mean (SD)	0.34 (±0.91)	0.81 (±1.23)	<0.001 *
Hypertension, *n* (%)	71 (75.5%)	241 (86.1%)	0.027 *
Diabetes mellitus, *n* (%)	22 (23.4%)	103 (36.8%)	0.024 *
Current smoking, *n* (%)	31 (33.0%)	47 (16.8%)	0.001 *
Alcohol consumption, *n* (%)	41 (43.6%)	117 (41.8%)	0.849
Anticoagulation, *n* (%)	7 (7.5%)	102 (36.4%)	<0.001 *
Clinical Characteristics			
Onset-to-door time, mean (SD)	587 (±1170)	584 (±1882)	0.991
NIHSS score, mean (SD)	5.4 (±4.5)	8.1 (±6.3)	<0.001 *
72hNIHSS score, mean (SD)	3.0 (±4.4)	6.5 (±6.3)	<0.001 *
Plasma glucose, mean (SD)	7.10 (±2.39)	7.52 (±2.61)	0.175
INR, mean (SD)	1.04 (±0.16)	1.21 (±0.63)	0.014 *
Treatment Modalities			
SC, *n* (%)	38 (40.4%)	133 (47.5%)	0.284
TL, *n* (%)	40 (42.6%)	54 (19.3%)	<0.001 *
MT, *n* (%)	11 (11.7%)	65 (23.2%)	0.024 *
TL + MT, *n* (%)	5 (5.3%)	28 (10.0%)	0.240
Imaging Characteristics			
LVO, *n* (%)	25 (26.6%)	142 (50.7%)	<0.001 *
ICA	1 (4.0%)	17 (12.0%)	0.316
MCA	19 (76.0%)	92 (64.8%)	0.360
M1	11 (44.0%)	50 (35.2%)	0.500
M2	5 (20.0%)	24 (16.9%)	0.775
M3	3 (12.0%)	18 (12.7%)	1.00
BA	1 (4.0%)	3 (2.1%)	0.481
VA	2 (8.0%)	11 (7.8%)	1.00
PCA (P1/2)	1 (4.0%)	9 (6.3%)	1.00
Tandem (ICA, MCA)	1 (4.0%)	5 (3.5%)	1.00
ACA (A1/2)	0 (0.0%)	5 (3.5%)	1.00
* CTP metrics, *n* (%)	26 (27.7%)	89 (31.8%)	0.519
rCBF < 30%, median (range)	0 (0–58)	5 (0–156)	-
Tmax > 6 s, median (range)	0 (0–202)	39 (0–343)	-
MMR, median (range)	0 (0–21)	3 (0–155)	-

Abbreviations: ESUS = embolic stroke of undetermined source, CE = cardioembolic stroke, SD = standard deviations, pre-mRS = pre-morbid modified Rankin Scale, NIHSS = National Institutes of Health Stroke Scale at admission, 72hNIHSS = National Institutes of Health Stroke Scale 72 h post-stroke, INR = international normalized ratio, SC = standard care, TL = thrombolysis, MT = mechanical thrombectomy, LVO = large vessel occlusion, ICA = internal carotid artery, MCA = middle cerebral artery, BA = basal artery, VA = vertebral artery, PCA = posterior cerebral artery, ACA = anterior cerebral artery, CTP = CT perfusion, rCBF = relative cerebral blood flow, Tmax = time-to-maximum delay, MMR = mis-match ratio (ratio of Tmax > 6 s to rCBF < 30%). * = CTP metrics were assessed in subset of patients. Due to limited sample size, no statistical comparisons were performed for these metrics.

**Table 2 neurolint-17-00035-t002:** Covariate balance before and after propensity score matching.

	SMD Before	*p*-Value	SMD After	*p*-Value
Baseline Characteristics				
Age	−1.148	<0.001 *	0.018	0.901
Sex	−0.035	0.770	0.130	0.377
Pre-mRS score	−0.438	<0.001 *	−0.265	0.072
Hypertension	−0.270	0.034 *	−0.243	0.100
Diabetes mellitus	−0.295	0.012 *	0.000	1.000
Current smoking	0.381	0.003 *	−0.023	0.878
Alcohol consumption	0.037	0.758	0.130	0.375
Clinical Characteristics				
NIHSS score	−0.482	<0.001 *	−0.328	0.027 *
72hNIHSS score	−0.514	<0.001 *	−0.092	0.530

Abbreviations: SMD = standardized mean difference, pre-mRS = pre-morbid modified Rankin Scale, NIHSS = National Institutes of Health Stroke Scale at admission, 72hNIHSS = National Institutes of Health Stroke Scale 72 h post-stroke. Note: * denotes statistical significance (*p* < 0.05)

**Table 3 neurolint-17-00035-t003:** Predictors of mRS-shift.

	Huber Coefficient	OR	95% CI	*p*-Value	Mean SHAPs Value	95% CI	Feature Importance
Baseline Characteristics							
Age	0.528	1.70	1.40–2.06	<0.001 *	0.373	0.320–0.425	15.16%
Sex	−0.047	0.95	0.80–1.21	0.595	0.016	0.013–0.019	0.66%
Pre-mRS score	−0.54	0.58	0.49–0.69	<0.001 *	0.255	0.194–0.316	10.36%
Hypertension	−0.039	0.96	0.81–1.14	0.655	0.013	0.010–0.016	0.52%
Diabetes mellitus	0.075	1.08	0.91–1.28	0.394	0.058	0.049–0.067	2.37%
Current smoking	0.165	1.18	0.99–1.41	0.078	0.021	0.014–0.028	0.85%
Alcohol consumption	−0.111	0.89	0.75–1.06	0.217	0.057	0.047–0.066	2.31%
Anticoagulation	0.112	1.12	0.94–1.34	0.213	0.064	0.052–0.075	2.59%
Clinical Characteristics							
Onset-to-door time	0.024	1.02	0.87–1.19	0.772	0.207	0.170–0.244	8.43%
NIHSS score	0.135	1.14	0.91–1.43	0.254	0.054	0.039–0.068	2.18%
72hNIHSS score	1.120	3.06	2.45–3.83	<0.001 *	1.206	1.065–1.347	49.05%
Plasma glucose	0.085	1.09	0.91–1.30	0.339	0.089	0.074–0.103	3.60%
INR	0.007	1.01	0.85–1.20	0.938	0.145	0.100–0.191	5.91%
Treatment Modalities							
SC	0.020	1.02	0.91–1.15	0.729	0.004	0.003–0.005	0.17%
TL	−0.084	0.92	0.81–1.06	0.189	0.014	0.012–0.017	0.58%
MT	0.127	1.14	1.00–1.29	0.050	0.006	0.005–0.008	0.25%
TL + MT	−0.087	0.92	0.79–1.06	0.237	0.001	0.000–0.002	0.05%

Abbreviations: mRS = modified Rankin Scale, OR = odds ratio, CI = confidence interval, SHAPs = SHapley Additive exPlanations, pre-mRS = pre-morbid modified Rankin Scale, NIHSS = National Institutes of Health Stroke Scale at admission, 72hNIHSS = National Institutes of Health Stroke Scale 72 h post-stroke, INR = international normalized ratio, SC = standard care, TL = thrombolysis, MT = mechanical thrombectomy. Note: * denotes statistical significance (*p* < 0.05)

## Data Availability

The original contributions presented in the study are included in the article, and further inquiries can be directed to the corresponding authors.

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
