# Peer review of "Shifting Outcomes: Superior Functional Recovery in Embolic Stroke of Undetermined Source Compared to Cardioembolic Stroke"

_2035-8377, 2025, doi:10.3390/neurolint17030035_

Round 1
Reviewer 1 Report
Comments and Suggestions for Authors
The study is well structured and well documented. The objectives, the methodology, and the results are clearly described.
However, I would like to note the following aspects:
1. I propose to revise the statement: "Comprehensive data were prospectively collected" bearing in mind that this is a retrospective study.
2. The introduction states: "The primary objective of this study is to compare long-term functional recovery between ESUS and EC patients, taking into account baseline differences."
The Limitations sub-chapter states, on the other hand: Furthermore, that the study's focus on 90-day outcomes may not fully capture long-term functional recovery, underscoring the need for longer follow-up to assess sustained recovery and stroke recurrence.
I would suggest using another suggestive term for the 90-day period.
Reviewer 2 Report
Comments and Suggestions for Authors
Dear Editor of Neurology International - MDPI
Thank you for inviting me into being a referee for this scientific article: ‘Shifting Outcomes: Superior Functional Recovery in Embolic Stroke of Undetermined Source Compared to Cardioembolic Stroke”.
Although I have transcribed my opinion of what is requested in the Medical Sciences - MDPI electronic form, I would like to add some comments.
- There are also some aspects that should be changed, which are referenced in red: (in red proposals to be potentially changed and also to be implemented).
Title: correct and nothing to propose.
Abstract
Well written, and provides a correct and adequate synthesis of the results found.
Introduction
Generally well written and with the necessary topics, well covered, but there are small aspects that need to be corrected:
Line 48: reads “RESPECT-ESUS, NAVIGATE-ESUS, ARCADIA and ATTICUS trials [9–12]; but according to the sequence of citations [9] is the NAVIGATE-ESUS study, [10] the RE-SPECT-ESUS study (not RESPECT-ESUS),
Thus the sequence should be NAVIGATE-ESUS [(rivaroxaban (at a daily dose of 15 mg) with aspirin (at a daily dose of 100 mg)], RE-SPECT-ESUS [Dabigatran 150 or 110 mg (for patients aged ≥75 years and/or with creatinine clearance 30 to <50 mL/minute) daily compared with aspirin 100 mg once daily], ARCADIA [Apixaban, 5 mg or 2.5 mg, twice daily (n = 507) vs aspirin, 81 mg, once daily (n = 508)] and ATTICUS [ apixaban (5 mg twice daily) compared with aspirin (100 mg once daily)].
Lines 65-66: reads “Therefore, the primary objective of this study is to compare long-term functional recovery between ESUS and CE patients while accounting for baseline differences”; but is 90-day a long-term assessment? Wouldn't it be better to specify 90 days rather than long term? In the article by Lee et al, citation [24] reads “1286 ± 748 days follow-up”…
Materials and Methods
Lines 83-84: reads “ESUS was defined based on the Cryptogenic Stroke/ESUS International Working Group criteria as a visible non-lacunar infarct in the absence of extracranial or intracranial”; The citation is missing and should be [13].
- Lined 112: reads “coagulation profile and vasculitis markers”; What markers of vasculitis were studied?
Results
The results are correctly and adequately presented, easy to read and understand (good flow-chart in Figure 1).
- Lines 198-199: reads “Additionally, fewer ESUS patients were on anticoagulation 198 therapy at the time of their stroke (13.5% vs. 36.4%, p = 0.005).”; Which anticoagulants and what proportion in each ESUS population vs CE patients, and what are the reasons for using anticoagulants? (especially patients with ESUS).
- Table 1. Row: Onset.door-time, means (SD) 360 (455); it must be 360 (± 455);
- Within the scope of the results there is one or two details that must be corrected and three important parameters are missing that can and should enrich this scientific work, and that should serve to elaborate a more extensive and enriching discussion (Table with 3 columns: presumed etiology; culprit arterial territory and stroke volume):
- a) Possible diagnoses of patients with ESUS are missing; It will be very useful to build a table (n = 37) (this aspect can be well grounded with the content of the Ntaios article (Ntaios, G. J Am Coll Cardiol. 2020;75(3):333–40. Embolic Stroke of Underminated Source): “The main pathologies that could be etiologically associated with embolic stroke of undetermined source (ESUS) could be broadly categorized in 7 embolic sources: atrial cardiopathy, covert atrial fibrillation, left ventricular disease, atherosclerotic plaques, patent foramen ovale, cardiac valvular disease, and cancer. For certain embolic sources like atrial cardiopathy and fibrillation, left ventricular disease, PFO, and cancer … the main pathophysiological stimulus for thrombogenesis is presumed to be the low blood flow, which predisposes to formation of red thrombi that may respond better to anticoagulation as sup ported by recent studies and meta-analyses (34,45,46)… On the contrary, in the case of atherosclerotic plaques in the aortic arch, cerebral, and intracranial arteries… the main pathophysiological trigger is plaque rupture and subsequent local platelet activation and aggregation leading to formation of white thrombi, which may respond better to antiplatelets….”
OR
In: Krutmann,S.;Wolf,C.;Aludin,S.; Larsen, N.; Seiler, A.; Schunk, D.; Jansen, O.; Seoudy, H.; et al. Cardiac CT in Large Vessel Occlusion Stroke for the Evaluation of Non-Thrombotic and Non-Atrial-Fibrillation-Related Embolic Causes. Neurol. Int. 2025, 17, 25. https://doi.org/10.3390/ neurolint17020025 “Cardioembolic aetiology was identified on cardiac CT in 211 cases (70%). After adjustment to AF and intracardial turbulence, multivariate regression analysis proves to be significantly associated with left ventricular dilation (adjusted chance rate (AOR) 32.4; 95% 3.0–349; p = 0.004), right interactive right (AOR 30.8; 95% IC 2.7–341.3; = 0.006) OR 24.5; 95% IC 2.2–270.9; P = 0.009), ATEROMA ARTICO> II (AOR 6.9; 95% IC 1.5-32.8; Bolic aetiology.
- b) Stroke location missing from the results of the current article: Arterial territory involved… occluded artery (ICA, M1 segment of MCA, M2 segment of MCA, tandem occlusion, or multivessel occlusion),…obviously this parameter will also have a different prognostic implication and is a variable that deserves to be discussed.
- c) Lack of territory volume/ischemic dimension, as can be seen, for example, In: Yoshimoto T, Inoue M, Tanaka K, et al. J NeuroIntervent Surg 2021;13:1081–1087 “…All MR DWI were retroactively postprocessed in an automated image postprocessing system (RAPID; iSchemaView Inc., Menlo Park, CA, USA). Ischemic core volume was calculated by the apparent diffusion coefficient…” Subjects were divided into three groups by baseline ischemic core volume (Group A: 70–100 mL, Group B: 101–130 mL, Group C: >130 mL),”… Estimates of the volume of the ischemic core and penumbral regions from CTP or diffusion and perfusion MRI scans…”; This topic (not addressed in this article) is of greater relevance in the discussion and interpretation of the results obtained.
- The 90-day assessment fails to describe the occurrence of AF (ECG/Holter monitoring at 90 days?).
- Table 2: Other legend SC = standard treatment, TL = thrombolysis, MT = mechanical thrombocytopenia - data not present.
- What anticoagulants did they take (CE)
o Lines 279-282
“3.3 Subgroup Analyses Anticoagulation Status
“Non-anticoagulated CE patients had a better functional recovery compared to anticoagulated CE patients (adjusted mRS-shift 2.09 ± 2.08 vs. 2.59 ± 2.30), although this difference was not statistically significant (p = 0.104)”; In this context it is important to say that if there are no statistically significant differences then there is no better functional recovery...this sentence needs to be changed; also the variation from 2.09 to 2.58 in a biological/clinical context has no meaning, and therefore the previous anticoagulation status has no impact on patients with Stroke ESUS or CE.
Lines 287 – 289: read “Across all treatment modalities, ESUS patients consistently demonstrated better functional recovery than CE patients, though the differences were not statistically significant….”; better functional recovery than CE patients, though the differences were not statistically significant? In this context, it is important to say that if there are no statistically significant differences then there is no better functional recovery (review this text).
Discussion
The discussion chapter has to be extended and improved. It is a short discussion (although everything written is scientifically correct) and the results obtained deserve a more extensive discussion, including the etiology of Strokes classified as ESUS and CE, the culprit territory and the Stroke volume as already mentioned. I really believe that the discussion of these variables in the impact on the results found in this scientific article is very important and will improve, from a practical clinical point of view, the repercussion of its conclusions.
- Lines 383-385: quote [17] reads “Strike features were predominantly ESUS (88%), within the first 2 years (75%), and involved a vascular territory other than the ESUS qualification (58%). Pre-existing cancer was the only independent predictor of recurrent stroke (adjusted hazard ratio [AHR] 5.43, 95% CI 1.43-20.64), recurrent ESUS (AHR 5.67, 95% CI 1.15-21.21), and higher mRS score at 3 months (ß 1.27, 95% CI 0.23-2.42)” which is somewhat different from what the authors write “…Consistent with existing literature, ESUS patients in our cohort were younger, had lower pre-stroke disability, fewer cardiovascular comorbidities, and experienced less severe strokes [5,16–18].”…the focus of the cited article is the involved arterial territory that is different and the cancer comorbidity.
- Line 390: reads “… which is within the range of previously reported rates of 17.5% [24] to 33.2% [25]”; but in the cited article [24] by Leee et al it reads “We classified ESUS into minor cardioembolic (CE) ESUS, arteriogenic ESUS, two or more causes ESUS, and no cause ESUS. Arteriogenic ESUS was subclassified into complex aortic plaque (CAP) ESUS and nonstenotic (<50%) relevant artery plaque (NAP) ESUS. A total of 775 patients were enrolled. During 1286 ± 748 days of follow-up, major adverse cardiovascular events (MACE) occurred (4.2 events/100 patient years). Among ESUS subtypes, CAP ESUS was associated with the highest MACE frequency (9.7/100 patient-years, p = 0.021). Cox regression analysis showed that CAP ESUS was associated with MACE (hazard ratio 2.466, 95% confidence interval 1.305–4.660) and any accident recurrence (hazard ratio 2.470, 95% confidence interval 1.108–5.508). The prognosis of ESUS varies according to the subtype, with CAP ESUS having the worst prognosis”; these aspects deserve to be integrated and compared with the data in the current article; also in the cited article it can be read “The rates of poor outcomes at 3 months were similar across ESUS subtypes (20.0% for CAP ESUS, 25.3% for NAP ESUS, 16.6% for minor CE ESUS, 14.1% for two or more causes ESUS, and 17.3% for no cause ESUS)…these data could be transcribed to the article under analysis….
- Lines 392- 398: reads “4.1 Impact of Age on Neuroplasticity and Recovery…”: there is not a single citation…there should be…discussing, for example, what can be read in the article on neuroplasticity by Xing, Y., Bai, Y. A Review of Exercise-Induced Neuroplasticity in Ischemic Stroke: Pathology and Mechanisms. Mol Neurobiol 57, 4 describes “the mechanisms by which exercise-induced neuroplasticity improves motor function and cognitive ability after stroke. The associated mechanisms include increases in neurotrophins, improvements in synaptic structure and function, the enhancement of interhemispheric connections, the promotion of neural regeneration, the acceleration of neural function reorganization, and the facilitation of compensation beyond the infarcted tissue.”
- Line 401-406: reads “4.2 Stroke Pathophysiology and Infarct Patterns” but once again this subchapter is very short…see the article on ESUS etiologies and outcomes….
4.4 Limitations and Future Directions and 5. Conclusions
Nothing to point out: they are correctly expressed.
References
Well-selected and representative literature on the topic; Suitable, but:
Citations should be revised in their format (correct the aspects highlighted in red:
- Feigin, VL; Brainin, M.; Norrving, B.; MARTINS, S. O.; Pandian, J.; Lindsay, P.; F Grupper, M.; Rautalin, I. World Stroke Organization: Global Stroke Fact Sheet 2025. International Journal of Stroke 2025, 20(2) 132 - 144, doi:10.1177/17474930241308142.; Missing 20(2) 132 – 144; the name of the publication should be: Int J Stroke.
- Ntaios, G.; Papavasileiou, V.; Lip, G.Y.H.; MILLIONIS, H.; Makaritsis, K.; Vemmou, A.; Koroboki, E.; MANIANOS, E.; Spengos, K.; Michel, P.; et al. Embolic Stroke of Undetermined Source and Detection of Atrial Fibrillation on Follow-Up: How Much Causality Is There? Journal of Stroke and Cerebrovascular Diseases 2016, 25, 2975–298; should be J Stroke Cerebrovasc Dis.
9- Hart, R.G.; Sharma, M.; Mundl, H.; Kasner, S. E.; Bangdiwala, SI; Berkowitz, SD; Swaminathan, B.; Washed, P.; Wang, Y.; 476 Wang, Y.; et al. Rivaroxaban for Stroke Prevention after Embolic Stroke of Undetermined Source. New England Journal of Medicine 2018, 378, 2191–2201; should be N Eng J Med.
- Cerebrovascular Diseases; must be Cerebrovasc Dis
- Journal of Neuroimaging; must be J Neuroimaging
- Scavasine, VC; RIBAS, G. da C.; COSTA, R. T.; Ceccato, G.H.W.; Zetola, V. de H.F.; Lange, MC Embolic Stroke of Undetermined Source (ESUS) and Stroke in Atrial Fibrillation Patients: Not so Different after All? International Journal of Cardiovascular Sciences 2021; should be Int J Cardiovasc Sci. 2021; 34(5):517-522
- Neurological Sciences; must be Neurol Sci.
In conclusion:
This is an excellent scientific work, well written and well explained. The data in this scientific work are correct but I believe there are some missing data that could enrich it and improve its scope (despite the small sample size); results should include data on the etiology of ESUS and CE stroke; This scientific work still suffers from an excessively short discussion that should be expanded as previously mentioned.

Reviewer 3 Report
Comments and Suggestions for Authors
Jessica Seetge and coauthors in the paper “Shifting outcomes: superior functional recovery in embolic stroke of undetermined source compared to cardioembolic stroke” retrospectively compared two groups of stroke patients, with embolic stroke of undetermined source (ESUS) and with cardioembolic stroke (CE). They focused on comparing the functional recovery of patients in these groups using a specialized modified Rankin scale (mRS). ESUS patients were younger, had lower pre-stroke mRS and underwent less severe stroke according to NIH Stroke Scale (NIHSS) in comparison with CE patients. So, it was not surprising that 90 days post-stroke mRS values were much lower (better recovery) in ESUS patients. However, the authors indicated that this difference was still valid even after adjusting for multiple differences in baseline patients’ characteristics. They also were trying to identify possible predictors of poor stroke outcomes in the analyzed patients. The paper is well written and illustrated and the data are thoroughly analyzed using modern statistical approaches. However, there are some serious remarks and questions concerning the novelty of the study and the obviousness and predictability of obtained results.
Major remarks and questions
1. The authors in the discussion section by themselves cited five papers [references 5, 19–22] with the close clinical results on comparison of ESUS and CE patients. In this regard, could they somehow clarify what principally new findings were reported in their own paper?
2. The authors compared the ESUS group with the CE group of patients with significantly more dangerous clinical condition. CE patients suffered a more severe stroke, they were older, they had higher pre-stroke mRS values and among them there were more patients with hypertension and diabetes. So, the results were rather predictable – patients in ESUS group recovered much better than in CE group (which in general confirmed previously reported observations – see point 1). In my opinion, it would be more informative not to take into account all numerous and significant differences between the groups using statistical analysis (multiple regression), but to compare the ESUS group not only with the total CE group but also with subgroup of CE patients with the close baseline clinical characteristics. Such a comparison will allow to more clearly identify the impact of possible pathophysiological differences in the development of ESUS and CE stroke on clinical outcomes.
Minor remarks and questions
1. Among 224 patients with cryptogenic stroke only 41 met the criteria of ESUS. Could the authors at least briefly explain what type of stroke was diagnosed in remaining 183 patients.
2. In baseline characteristics the authors do not mention previous stroke or TIA episodes. Does it mean that there were no such patients in both groups?
3. In baseline characteristics the authors do not mention the antiplatelet therapy. Does it mean that all patients were free of any antiaggregants?
4. Do the authors measure the D-dimer in their patients? It would help to assess the thrombotic burden ant its impact on the difference between analyzed groups as well as on clinical outcomes
5. It is not indicated anywhere in which group the predictors of clinical outcomes were analyzed (see section 3.4). It should be clearly indicated in the text, as well as in Table 2 and Figures 4, 5. The results of this large part of the study are not mentioned (for unknown reason) in the “Abstract” or even in “Discussion” section.
6. The authors use rather sophisticated statistical analysis in assessing possible predictors of clinical outcomes. Could the authors presented statistical results in a slightly more simple way or with the more thorough explanations for the readers, who are not yet well familiar with these approaches (like “Firth regression analysis and XGBoost with hyperparameter tuning and k-fold cross-validation” or “Beeswarm plot of mean SHAP (SHapley Additive exPlanations) values”). Probably just briefly indicate the meanings and advantages of these tests.
7. Figure 4. It is better to indicate in the legend which points are marked with red crosses
Round 2
Reviewer 3 Report
Comments and Suggestions for Authors
The authors significantly revised the manuscript in accordance with the comments made in the initial review. After a more thorough assessment of the diagnostic criteria, they almost tripled the group of ESUS patients, which made it possible to perform statistically correct comparison of this group not only with the total CE group, but also with the CE subgroup adjusted for baseline characteristics. This significantly strengthened the results of the study. Nevertheless, there are still some remarks concerning the presentation of the material.
- I failed to find the reference to Table 1 in the text
- The authors stated “The study cohort consisted of 172 male patients (46.0%)… (line 233). It is more correct to state that the study cohort consisted of 374 patients and then indicate how many of them were male.
- The authors remained in the revised manuscript figures from the initial version and in my variant the initial figure 3 “Density plot of adjusted mRS-shift” (presumably deleted) hidden the half a page of corrected text that could not be analyzed (lines 315-331)
- The authors still do not answer why they do not mentioned in the “Abstract” or even in “Discussion” section the large part of “Results” addressing the evaluation of “Predictors of Functional Recovery” (Paragraph 3.7) (see remark 7 in initial review)
- In response to comment No 3, the authors indicated that among the group with cryptogenic stroke, 16 patients were eventually diagnosed as with CE stroke. It is unclear whether these patients were transferred to the appropriate group or not.
